# Preparation of Cemented Carbide and Study of Copper-Accelerated Salt Spray Corrosion and Erosion Behavior

**DOI:** 10.3390/ma15197023

**Published:** 2022-10-10

**Authors:** Shasha Wei, Yuanyou Li, Renxin Wang, Hu Yang, Ziming Guo, Rongchuan Lin, Qingmin Huang, Yuhui Zhou

**Affiliations:** College of Marine Equipment and Mechanical Engineering, Jimei University, Xiamen 361021, China

**Keywords:** tungsten–cobalt carbide, powder sintering, acidic copper-accelerated salt spray corrosion, erosion behavior

## Abstract

(1) Mud pulser carbide rotors, as a core component of ground communication in crude oil exploration, are often subjected to mud erosion and acid corrosion, resulting in pitting pits on the surface, which affects the accuracy. The purpose of this study was to investigate the acid corrosion and erosion behavior of cemented carbide materials and provide a reference for the wider application of cemented carbide materials in the petrochemical industry. (2) Experimental samples of tungsten–cobalt carbide were sintered at a low pressure by powder metallurgy. The petrochemical application environment was simulated by accelerated salt spray corrosion and solid slurry erosion with the aid of acidic copper, and the experimental phenomena were analyzed by SEM (scanning electron microscope), EDS (Energy Dispersive Spectroscopy), and XRD (X-ray diffraction). (3) The experimental results show that the coercivity of the pitted cobalt-cemented tungsten carbide prepared in this study was 17.89 KA/m, and the magnetic saturation strength was 14.42 G·cm^3^/g. The corrosion rate was the fastest during the acidic copper acceleration experiments from 4 h to 16 h, and the corrosion products of WCo_3_ and Co_3_O_4_ were generated on the corrosion surface. The maximum erosion rate of 0.00104 in the erosion experiment corresponds to a corrosion sample with a corrosion time of 36 h. (4) Therefore, the coercive magnetic force and magnetic saturation strength could be derived from the prepared carbide hard phase grains and carbon content in the appropriate range. The corrosion product in the corrosion process slowed the corrosion rate, and a large amount of cobalt and a small amount of tungsten was lost by oxidation during the corrosion process. The corrosion time had the greatest effect on the erosion performance of the carbide, and the long corrosion time led to surface sparseness, which reduced the erosion resistance.

## 1. Introduction

Crude oil is called the “blood of industry” and is the raw material for many chemical products, such as petroleum, fertilizers, pesticides, lubricants, etc. [1]. The key component of the mud pulser (measurement while drilling, MWD) is a pulse rotor made of carbide material, as shown in Figure 1. However, the small axial clearance during operation makes the rotor more susceptible to erosion and corrosive wear, which reduces the quality of signal transmission [2,3,4,5,6,7]. Mohammad, E. et al., studied the erosion rate of rotor cemented carbide material in deep wells using finite element simulation and verified it experimentally [8]. Zhang, W. et al., also investigated the erosion wear mechanism of cemented carbide material for rotors in mud containing solid phase particles and studied the effect of different erosion angles and abrasive sizes on the erosion wear performance of cemented carbide [2]. WC-Co cemented carbide is widely used in military, aerospace, machining, metallurgy, oil drilling, mining tools, and other fields, such as cutting tools, rotors, and molds, because of its excellent mechanical properties, such as high hardness, high wear resistance, and high toughness [9,10]. However, in the preparation of ultrafine grain cemented carbide to improve the alloy properties, WC-Co composite powder is the key factor [11,12]. Zhang, F.L. et al., prepared WC-Co cemented carbide material using the ball milling method and studied its microstructure. It was found that ball milling can effectively refine the microstructure of WC, and the grain size was significantly reduced, but it was easy to introduce internal strain and cause dislocation defects in the material [13]. Bonache, V. et al., made tungsten carbide powder using the freeze-drying method in an aqueous solution and characterized the density, microstructure, and fracture toughness of the powder and the sintered material. Studies have shown a uniform increase in the hardness and toughness of the sintered material, as well as the ability to control grain growth and develop materials with comprehensive mechanical properties through this technique [14]. In addition, acid corrosion has been also a major problem faced by the oil and gas industry. The current research on the mechanism of acidic corrosion has mainly included the influence of acidic corrosion media, metal materials, organization, and other key parameters [15]. Wu Zhijian took cemented carbide prepared by the traditional powder metallurgy process and investigated the effects of Ni and Cr on corrosion behavior using acidic salt spray corrosion experiments [16,17]. Human, A.M. et al., investigated the corrosion behavior of pure WC and WC-Co cemented carbide in industrial applications [18]. Guo, S. et al., studied the corrosion resistance and corrosion micromorphology of WC-Co cemented carbide in an acidic environment [19]. The petrochemical environment also contains a large number of erosion phenomena, which can be considered pitting fatigue damage to the surface of metallic materials caused by the high-speed flow of tiny solid particles in liquids. WC-Co cemented carbide is commonly used as an erosion-resistant material, and Beste, U. et al., investigated the effect of different erosion particle sizes on the erosion performance of tungsten–cobalt cemented carbide, including the erosion rate and erosion micromorphology [20]. Wang, G. et al., deposited diamond coatings on cemented carbide surfaces to improve the resistance to cement slurry jet erosion [21].

In summary, the research on cemented carbide materials has focused on the material preparation and microscopic characterization of cemented carbide and the investigation of corrosion and erosion properties. Crude oil is rich in C, H, and small amounts of S and N, which easily form an acidic environment. It also contains solid phase particles of mud fluid, which is liable to produce the erosion phenomenon. The existing cemented carbide materials for petrochemical environments only consider the effect of solid phase particles on the erosion and lack the influence of acidic corrosion on erosion performance. Therefore, this study used alcohol ball mill powder and low-pressure sintering to prepare WC-Co cemented carbide for petrochemical environments, which is rich in rare earth elements. We used an acidic copper-accelerated salt spray corrosion test and ablation test to simulate the environment with both acidic corrosion and solid phase particle ablation and investigate the acidic corrosion and high-pressure slurry erosion behavior of WC-Co cemented carbide material for MWD rotors in a petrochemical environment. We performed a comprehensive simulation of the environment of the cemented carbide materials so as to study the corrosion mechanism of cemented carbide and corrosion mechanism of cemented carbide and explore the comprehensive performance of corrosion resistance and erosion resistance of the prepared cemented carbide. The comprehensive properties of the cemented carbide were tested using magnetic saturation and coercive force. The magnetic saturation strength was affected by the chemical composition of the cemented carbide, but not by the microstructure. The coercive magnetic force was sensitive to the chemical composition and microstructure of the cemented carbide. It is of great significance to control the comprehensive properties of cemented carbide by the constraint of coercive force and magnetic saturation value.

## 2. Experimental Design

### 2.1. WC-Co Cemented Carbide Sample Preparation

WC-Co cemented carbide samples were prepared using the powder metallurgy method; the specific preparations were as follows: WC and Co powders with particle sizes of 3.5 μm and 1 μm, respectively, were used. A carbide spherical grinding body with high hardness was placed in the horizontal rolling ball mill cavity, while the prepared powder and additives, including paraffin and industrial alcohol, were loaded into the ball mill cavity. The ratio of the ball to material was 4:1, the speed of the ball mill was 36 r/min, and the slurry was ground for 72 h. The carbide was ground to the required particle size and mixed with Co powder to achieve good material uniformity and density. The role of the paraffin wax was to increase the plasticity, staining, and lubrication of the mixture and prevent oxidation and easy granulation. The alcohol, as the wet grinding medium, reduced the oxygen content in the mixture, thus reducing the oxygen content in the sintering process and obtaining better sintering performance.

Then, the ground mixed slurry was pumped into a drying tower, atomized, and dried at an inlet temperature of 180 °C and an outlet temperature of 95 °C. The cyclone was recycled into a mixed dry fine powder. The chemical composition of the mixed powder is shown in Table 1. The Hall flow rate was used to measure the time it took every 50 g of mixed material to pass through the small hole of the Hall flow meter, and the Hall flow rate of the powder was measured as 40.7 s/50 g.

The mixed powder was pressed into round bars using a 25-ton cantilever hydraulic press, as shown in Figure 2a. Subsequently, a low-pressure integrated sintering furnace (Xin da GN-5518-6MPa-6, Xinda Powder Metallurgy Equipment Manufacturing Co. LTD, XiangTan, China) was used for pre-sintering and dewaxing, and the sintered products were vacuum sintered and held for 60 min at 1400 °C. The sintered products are shown in Figure 2b below. The chemical composition of the WC-Co cemented carbide round bar sample, with a size of h = 12 mm, R = 4 mm, is shown in Table 2.

### 2.2. Corrosion and Erosion Experiments

Before the experiment, the sample needed to be ground and polished to achieve a mirror effect, with a maximum surface roughness of 0.12 μm, and ultrasonic cleaning to remove surface stains. The bottom of the cylindrical test piece was dipped in a plastic plane, and then the whole piece was placed in the equipment to ensure that each polished experimental end face was exposed to salt spray at 20° to the vertical direction, as shown in Figure 3a. Anode: The metal entered the solution as a hydrated ion and left an equivalent amount of electrons in the metal, such as Co^2+^. Cathode: The remaining e^−^ in the metal was depolarized by oxygen and reached the surface of the cathode by diffusion or convection, absorbing electrons and becoming hydroxide ions (OH^−^). Then, the Co^2+^ and WC chemical reaction generated corrosion products Co_3_O_4_, WCo_3_, and CO_2_, resulting in the loss of Co in the carbide surface crater formation, as shown in Figure 3b. In the acidic copper-accelerated salt spray test, the salt solution with a small amount of CuCl strongly induced corrosion and was a rapid salt spray corrosion test. The test was conducted in accordance with the “GB/T 10125-2021 experimental standards”, and the solution had a concentration of 50 g/L ± 5 g/L of NaCl salt solution, ice acetic acid, and a concentration of 0.33 g/L ± 0.02 g/L of copper chloride solution. The other test parameters are shown in Table 3. The experimental equipment model was the DCTC1200P salt spray chamber (Wanda Experimental Instrument Co. LTD, ChongQing, China), and the longest corrosion time of the specimen was 36 h, with continuous spray. There were three groups of samples and corrosion times of Y1 (12 h), Y2 (24 h), and Y3 (36 h), and the groups of two samples each were recorded as Y11, Y12, Y21, Y22, Y31, and Y32. Salt spray corrosion sample cleaning and drying: The samples were submerged in pure water and underwent ultrasonic cleaning for 10 min. Then, the pure water was replaced with anhydrous alcohol as the medium for ultrasonic cleaning for 10 min. The samples were saturated in anhydrous alcohol and then left on filter paper to dry. During the test, the dried corrosion samples were weighed at certain intervals on a 10,000 ppm analytical balance METTLER TOLEDO (ME204) (METTLER TOLEDO, Zurich, Switzerland).

According to the information about the deep well environment, it is known that the solid phase component of the mud is mainly quartz sand [22,23]. The specimens were weighed for mass before erosion (mb), and the samples were fixed on the type table at an angle before the experiment. The rest of the erosion parameters are shown in Table 4. Using the erosion equipment, model YQ-ZP3W (Yue Qiang Machinery Technology Dong Guan, China) measured the test equipment (manufactured according to the national standard ASTM-G76-83). The erosion time was 10 min, and the specimens were weighed again to record the mass after erosion (ma). Then a high-magnification microscope (LEICADM2700 M, Leica Microsystems, Wetzlar, Germany) was used to observe the surface condition of the carbide after etching.

Before and after the experiments, the samples were observed by scanning electron microscopy (FE-SEM, ZEISS, XiaMen, China) with energy dispersive X-ray spectroscopy (EDS) to observe the microscopic morphology as well as the energy spectrum and analyze the elemental composition of the corrosion products and the XRD crystallographic corrosion products.

## 3. Results

### 3.1. Cemented Carbide Samples

#### 3.1.1. Microstructure of Cemented Carbide Powder Raw Materials

The quality of the powder mixture determines the quality of the cemented carbide, and a stable and homogeneous mixture is required to prepare a cemented carbide with excellent properties. Figure 4a shows the morphology of the dried carbide powder particles. The average particle size of the powder was less than 120 μm. It was mixed with WC, Co, and another rare earth element, Ta. Alcohol diluent was added and mixed into a slurry, and the powder was gradually refined and mixed after a long time of ball milling. The mixed slurry formed small droplets in the centrifugal atomization drying process and gradually shrunk to a spherical shape under the action of the surface tension of the droplets. During the spraying process, the water on the surface of the spherical droplets continuously evaporated and was lost by the hot airflow, and the solute part crystallized on the surface of the droplets to form a solid. The internal liquid was also continuously lost to the outside, thus forming a hollow spherical structure. Many researchers have also studied the grinding of mixed powders into spherical particles [24,25,26,27,28]. Figure 4b shows the local magnified morphology of the hollow spheres, which shows the obvious hollow gaps. The hollow interstitial structure led to a significantly larger contact area of Co, which introduced more chemo-oxygen, which was involved in the carbon–oxygen reaction during the sintering and directly affected the carbon content in the alloy. As can be seen in the figure, apart from the individual powder grains aggregated up to 1μm in size, the rest of the grains were extremely fine (≤0.7 μm) and uniform in size, and all the WC grains were covered by Co phase. There was also Co between the grains, forming a stable hollow structure. From the microscopic morphology, it can be seen that except for a very small number of shaped powder grains, they were distributed into a uniform spherical shape, and the surface of the sphere was relatively smooth. The more rounded the grains and the smoother the surface, the better the flowability of the powder mixture, and the more stable the molding is in the high-speed pressing process [29,30,31].

#### 3.1.2. Structure of Cemented Carbide Specimens

The properties of cemented carbide depend on its composition and structure, and once the composition is determined, it is necessary to consider its structure. In this paper, the structure of this cemented carbide sample was studied in terms of the alloy phase composition, grain size, and tissue defects. The relative magnetic saturation value of the alloy was used to measure the carbon content of the alloy and the coercivity magnetism to measure the grain size of the alloy.

The elemental content of the uncorroded sample was analyzed using scanning electron microscopy (FE-SEM, ZEISS) with energy dispersive X-ray spectroscopy (EDS) on the surface of the sample, and Figure 5a shows the elemental content obtained by the EDS surface scanning. From this, we found that the metal element content of the cemented carbide sample did not differ much from the element content of the mixed powder (Table 1). A small amount of added Ta refined the WC grains, but too much Ta led to a decrease in the alkali corrosion resistance and toughness of the alloy. A very small amount of the element Tb was added to improve the coercivity during the sintering. The elements W and C form the main alloy component WC, which is used to regulate the cobalt phase (γ phase) composition, crystalline shape, and grain size of the alloy [32,33,34]. Figure 5a shows the scanning electron microscope (FE-SEM, ZEISS) secondary electron mode scan of the carbide obtained after etching with solution (Murakami’s). It mainly contains the WC and cobalt phases of the alloy. In the figure, it can be observed that a few WC grains were overgrown, which led to oversized grains and affected the crystal homogeneity. The additive Ta suppressed grain growth. Cobalt phase is a Co-like solid solution of WC and W, and it can be seen that WC is surrounded by the cobalt phase. In other words, the WC also surrounds the cobalt phase, which enhances the dip strength between the grains. Figure 5c shows a metallographic image of the surface of the alloy sample after polishing and etching using a high magnification optical microscope (LEICA DM2700M) at 500×. The larger WC plate crystal structure can be seen in the figure, and the yellow area is the corresponding Co dip phase, which indicates that the Co distribution was uniform and there was no Co enrichment, decarburization, or other defects. However, some of the WC particles were still too large, as mentioned above. The WC crystal plate structure was irregularly distributed, so it produced more voids, and the Co phase filled the voids, which may have led to Co enrichment and affected the comprehensive performance of the alloy. The coercive magnetic force (HC) was 17.89 kA/m as measured by the Changsha Zhongda ZDHC40 coercive magnetic force automatic measuring precision instrument (Chang sha Zhongda Precision Instrument Co. LTD, Chang Sha, China). The magnetic saturation strength was 14.42 G·cm^3^/g < 15.98 G·cm^3^/g as measured by the Changsha Zhongda ZDMA6535 specific saturation magnetization strength measuring precision instrument (Chang sha Zhongda Precision Instrument Co. LTD, Chang Sha, China), which indicates that the magnetic saturation requirement of this YG type cemented carbide was met, and the alloy properties were also within the specified range. S. Sundin’s formula is as follows:PH_c_·M_s_ ∝ S_WC_/S_Co_
(1)

H_c_: the coercive magnetic force of the alloy;

M_s_: the saturation magnetization strength of the alloy;

SWC/S_Co_: the interface area between the Co and the WC in the alloy, i.e., the surface area of WC grains in the alloy.

According to Equation (1), the coarser the WC grains and the smaller the surface area of the WC grains, the thicker the Co layer that can be analyzed for the same Co content. The coercive magnetism is related to the Co content, sintering time and process, added elements, and impurities. The measured coercive magnetism value was 17.89 kA/m < 20 kA/m.

**Figure 5 materials-15-07023-f005:**
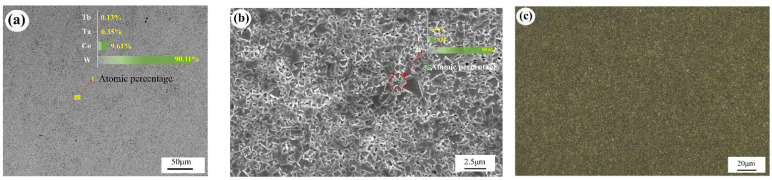
(**a**) Surface element distribution of the sample, (**b**) SEM image of the cemented carbide after etching, (**c**) micro-metallographic image of the cemented carbide sample.

### 3.2. Sample Corrosion Performance Analysis

#### 3.2.1. Corrosion Morphology and Rate

Figure 6 shows the macroscopic morphology of the sample surface at 100 and 500 times magnification under different acidic copper-accelerated salt spray corrosion times taken by high magnification optical microscope (LEICA DM2700M). Figure 6a shows the surface condition of the uncorroded sample with a smooth surface and no obvious mottling or delamination. Figure 6b–d shows the corrosion after 12 h, 24 h, and 36 h on the surface of the sample. The surface appears to have different depths and sizes of rust spots and corrosion holes, and the entire surface shows grooves and delamination, indicating that a large amount of elements was lost during the corrosion process. Additionally, the surface corrosion after the generation of a large number of WCo compounds at 24 h and 36 h appears to have different depths of color. Cobalt oxides with different oxygen-containing elements (gray-green for CoO and black for Co_3_O_4_ [27]) appeared with the deepening of corrosion, which led to various color differences on the surface of the samples. Co, the bonding phase in tungsten–cobalt carbide, is a corrosive material and easily soluble in various acidic substances, such as glacial acetic acid. It is extremely soluble in alkaline substances. The flow of acidic salt spray (a mixture of glacial acetic acid, copper chloride, and sodium chloride) and the corrosive loss of Co in the carbide during the corrosion process led to rust spots and holes in the corrosion etching. Sparse corrosion holes appeared after 12 h of corrosion, significant corrosion holes appeared after 24 h of corrosion, and corrosion gullies appeared after 36 h of corrosion.
(2)$$V_{\mathrm{corrosion}}=\frac{m_{0\;-\;}m_{1}}{\mathrm{st}}$$

V: metal corrosion rate, mg·m^−2^·h^−1^;

m_0_: weight of metal before corrosion;

m_1_: weight of metal after corrosion, as shown in Table 5;

s: area of the corrosion surface of the metal sample;

t: corrosion time.

**Figure 6 materials-15-07023-f006:**
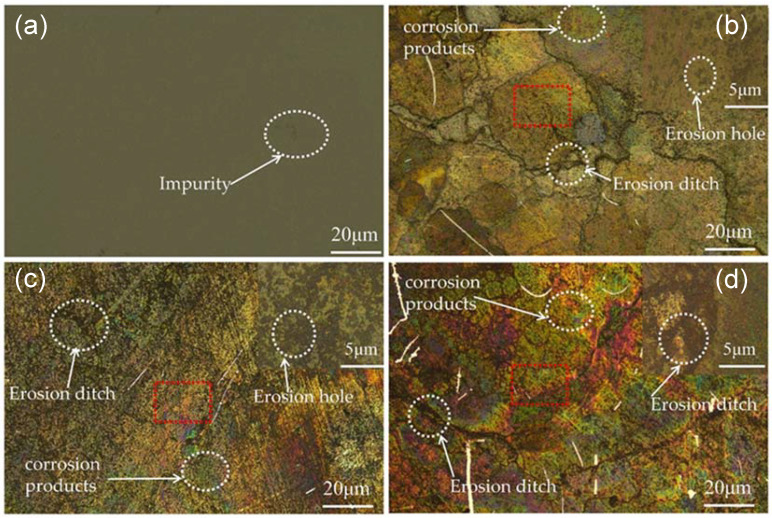
(**a**) Uncorroded sample surface, (**b**) sample surface after 12 h of corrosion, (**c**) sample surface after 24 h of corrosion, (**d**) sample surface after 36 h of corrosion.

**Table 5 materials-15-07023-t005:** Sample mass before and after corrosion.

Sample	m_0_ (g)	m_1_ (g)	Corrosion Time (h)
Y33	8.795	8.7949	4
Y33	8.795	8.7946	8
Y33	8.795	8.7940	12
Y33	8.795	8.7933	16
Y33	8.795	8.7926	20
Y33	8.795	8.7919	24
Y33	8.795	8.7913	28
Y33	8.795	8.7907	32
Y33	8.795	8.7901	36

Note: m_0_: weight before corrosion, m_1_: weight after corrosion.

Through the weighing determination method in accordance with Formula (1), the corrosion rate of the salt spray corrosion process of cemented carbide samples can be calculated, and the change curve is shown in Figure 7. The corrosion rate shows that the corrosion rate of the 4–16 h time period was the fastest, and the corrosion rate of the 16–36 h time period gradually declined, mainly because the corrosion process generated for the continuation of corrosion had a certain protective effect in addition to covering the surface of the corrosion generated by reducing the contact with oxygen, thereby slowing down the corrosion rate.

m_0_ represents the weight of sample Y33 before corrosion, that is, the corrosion time of the third group of samples was up to 36 h. m_1_ is the real-time weight of the samples every 4 h, representing the weight change during the corrosion process. The reason there was only one Y33 is because it was the one that corroded the longest so the weight data are adequate. Figure 7 is a bar chart with error bars based on the data in Table 5.

#### 3.2.2. SEM and EDS Analysis of Corroded WC-Co Surface

The surface SEM morphology of cemented carbide under different corrosion cases is shown in Figure 8, and the corrosion products were mainly composed of W, Co, and O, as shown by the EDS analysis results. In Figure 8a, after corrosion, a large number of granular corrosion oxides were sparsely distributed on the carbide sample surface, and there were still many exposed WC grains with slight corrosion and thin oxide skin thickness according to local magnification. Figure 8b corrosion particles appeared to be connected and the number of particles increased, but the increase rate did not increase with the corrosion time. Figure 8c shows that the oxide particles were connected, gradually forming sheets. The thickness increased, the O and Co started to increase, and the corrosion effect was intense. With the increase in corrosion time, the sample surface corrosion products gradually transitioned from sparse granulation to dense flakes, and the thickness of the corrosion oxide seen in the morphology of the figure also gradually increased. A high amount of oxygen in the salt spray corrosion process gradually reacted with the Co and W, generating an oxide skin. In the corrosion process, the sample surface W almost disappeared, while the surface of the product Co and oxygen increased, indicating that the corrosion process Co was consumed by oxidation, and the W oxidation corrosion was less.

#### 3.2.3. XRD Analysis

XRD analysis was performed on the carbide surface after salt spray corrosion for 0 h, 12 h, 24 h, and 36 h, and the results are shown in Figure 9. The characteristic peaks appearing at 2θ around 31.5°, 35.5°, 48°, 64°, 66°, 73°, 75°, 77°, and 84° point to the substance WC, and the corresponding PDF number is 04-003-7145 (PDF# 04-003-7145). The characteristic peaks at 2θ around 31.5°, 35.5°, 44°, 48°, 64°, 66°, 73°, 75°, and 84° point to the substance WCo3 (PDF#04-004-2222). The characteristic peaks appearing at 2θ around 31.5°, 35.5°, 44°, 66°, and 77° point to the substance Co_3_O_4_ (PDF#98-000-0166), which corresponds to the Co oxides formed after the enrichment of the large amount of Co and O that appeared in the EDS analysis. The rest of the peaks were not retrieved with matching phases and were considered to be a small amount of adulterated material or error-induced spurious peaks. The XRD analysis results show that the main material components of the sample proved to be WC, WCo_3_, and Co_3_O_4_. The results indicate that a large amount of Co was lost during the corrosion to form corrosion products with O and W, resulting in a large loss of Co and a small amount of W.

### 3.3. Erosion Corrosion Behavior of WC-Co Materials

#### 3.3.1. Erosion Morphology and Rate

During the erosion process, a well-mixed abrasive fluid (a mixture of abrasive and water) was delivered to the gun by the abrasive pump. In addition, compressed air was used as an additional acceleration power for the abrasive fluid, which passed into the gun through the air delivery pipe. Inside the gun, the compressed air accelerated the abrasive fluid entering the gun and ejected it at a high speed through the nozzle, spraying the sample surface according to the set angle for the analysis of the erosion resistance of the sample. In Figure 10, it can be seen that the degree of salt spray corrosion had a great influence on the erosion resistance of the cemented carbide. Figure 10a shows the uncorroded carbide surface, and no obvious erosion pits or scratches appeared after erosion. Figure 10a–c shows the surface of the carbide samples corroded for 12 h, 24 h, and 36 h in turn. With the increase in corrosion time, the diameter of the erosion pits gradually increased, and they all showed more regular spherical pits. In addition, a smooth surface around the erosion crater appeared, and the depth of the crater gradually increased with the increase in corrosion. In practical application scenarios, the presence of erosion pits on the surface of rotor workpieces will aggravate further corrosion and affect the accuracy and life of the instrument. The effect of corrosion on the erosion resistance of the cemented carbide was further characterized by recording the mass before and after erosion and calculating the erosion rate of the cemented carbide using Equation (2) [19], as shown in Figure 11.
(3)$$\Delta M=\frac{m_{b\;}{-\;m}_{a}}{m_{b}}$$

ΔM: the rate of change of mass during metal erosion;

m_b_: metal weight before erosion;

m_a_: metal weight after erosion, see Table 6.

**Figure 10 materials-15-07023-f010:**
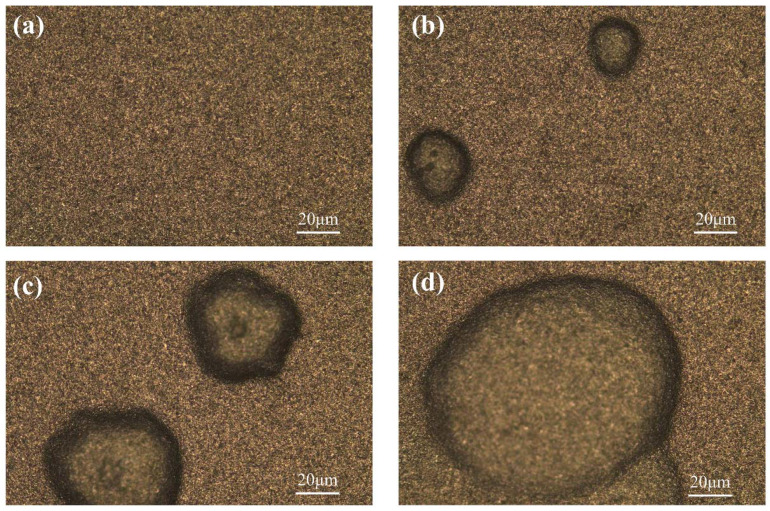
(**a**) Uncorroded sample erosion surface; sample erosion surface after (**b**) 12 h, (**c**) 24 h, and (**d**) 36 h of corrosion.

**Figure 11 materials-15-07023-f011:**
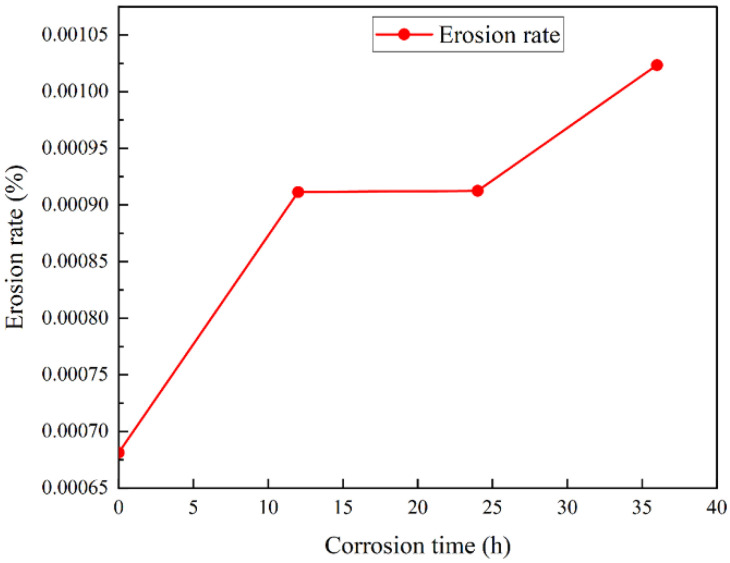
Erosion rate change curve during the erosion process.

In Figure 11, it can be seen that the salt spray corrosion had a huge impact on the erosion resistance of the cemented carbide (uncorroded erosion rate of 0.00068; corrosion rate of 0.00104 after 36 h). The erosion rate increased gradually with the increase in the salt spray corrosion time, for 12 h and 24 h of corrosion. The erosion rate is related to the time period with the decrease in the corrosion rate. As the corrosion time continued to increase, the generation of corrosion increased and the depth deepened, resulting in a significant increase in the erosion rate.

#### 3.3.2. SEM Analysis of the Erosion Morphology

During the experiment, the SiO_2_ particles hit the surface of the corroded sample at an oblique high speed, forming irregular pits and scratches. For the uncorroded carbide surface after erosion morphology, as shown in Figure 12a, the surface scratches were uniform long stripes, corresponding to 60° SiO_2_ particles across the surface. In addition, the surface properties of the uncorroded carbide were more uniform, thus forming uniform scratches. The right side shows the magnified surface morphology, and it can be seen that the uncorroded surface was relatively smooth during the erosion process and there were few erosion pits and surface flaking marks. However, after slight corrosion (Figure 12b,c), the length of the scratches became shorter and uneven, corresponding to the difference in surface properties after corrosion. A large number of scour pits and a small number of peeling marks appeared. With further corrosion (Figure 12d), there were almost no long scratches on the surface after erosion, and they were plastic scratches with a messy distribution. A large loss of Co produced loose oxides on the surface, and the oxides did not have enough resistance to the erosion process, resulting in deep erosion pits and particles on the raised surface.

## 4. Conclusions

By performing acidic copper-accelerated salt spray corrosion on the prepared pitted cobalt carbide, the following conclusions can be drawn from the erosion and corrosion experiments at different levels of corrosion.

(1)The research and quality control of the mixed powder used for the sintering enabled the sintering of the carbide samples with a carbon content and grain size that met the requirements, and the measured coercive magnetic force and magnetic saturation strength were 17.89 KA/m and 14.42 G·cm^3^/g, respectively, meeting the carbide production standards.(2)The acidic copper-accelerated salt spray corrosion experiments on cemented carbide showed that the corrosion rate was different with different corrosion time periods. The corrosion rate was fast at the 4–16 h stage, and the generation of corrosion in the corrosion process slowed down the corrosion rate. A large number of rust spots and corrosion holes appeared on the surface during the corrosion process and generated corrosion products of WCo_3_ and Co_3_O_4_, which caused a large amount of cobalt and a small amount of tungsten oxidation loss.(3)After corrosion, the erosion experiments show that the corrosion of the carbide had a great impact on the anti-corrosion performance. The longer the corrosion time was, the greater the erosion rate was, reaching 0.00104. The long corrosion time led to surface loosening, which reduced the erosion resistance. Erosion occurred after a large number of erosion pits formed, and surface flaking seriously affected the durability and accuracy of the material.

## Figures and Tables

**Figure 1 materials-15-07023-f001:**
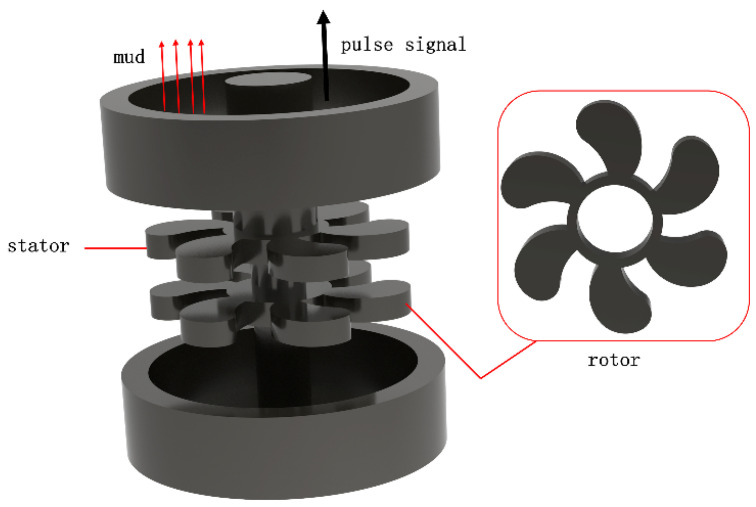
Schematic diagram of MWD pulser and rotor.

**Figure 2 materials-15-07023-f002:**
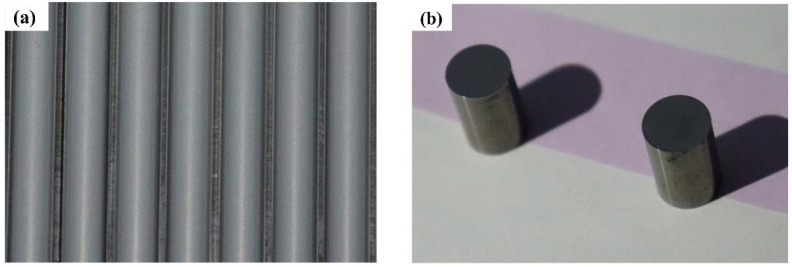
(**a**) Powder pressed bar before sintering, (**b**) cemented carbide sample bar after sintering.

**Figure 3 materials-15-07023-f003:**
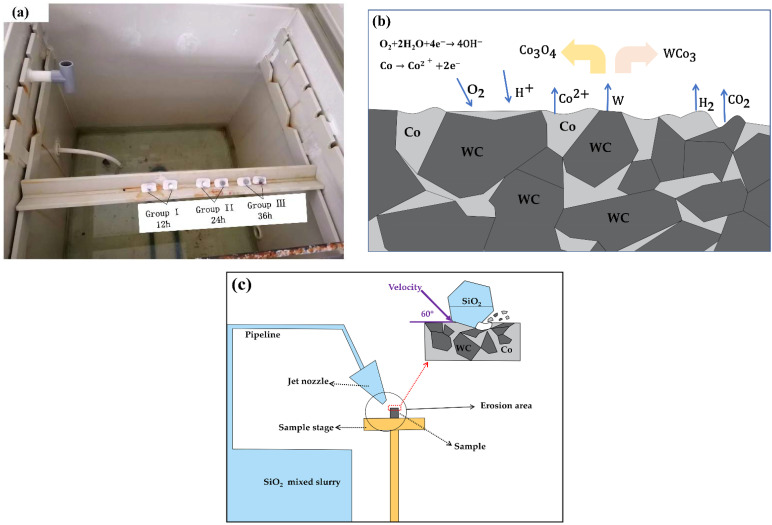
(**a**) Salt spray test sample placement diagram, (**b**) basic principle of salt spray corrosion process of cemented carbide, (**c**) schematic diagram of erosion.

**Figure 4 materials-15-07023-f004:**
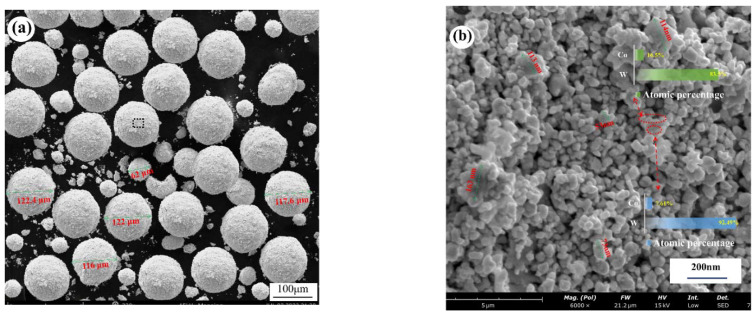
(**a**) Microscopic morphology of mixed powder particles, (**b**) morphology of mixed powder particle distribution.

**Figure 7 materials-15-07023-f007:**
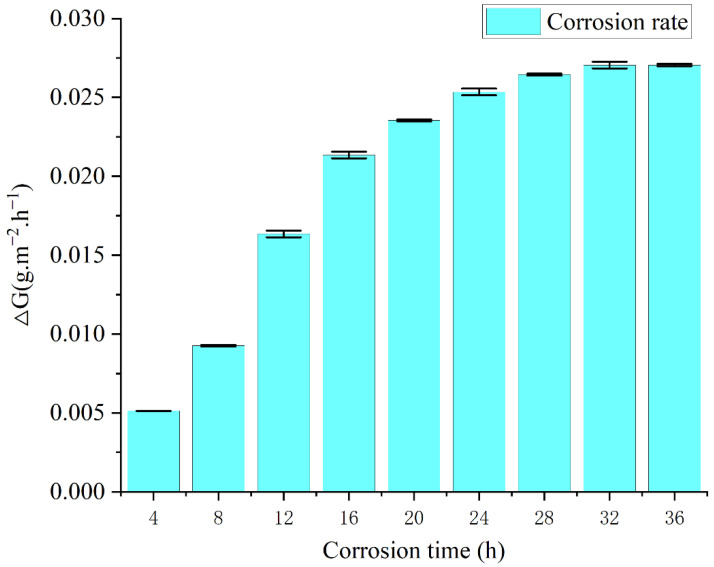
Corrosion rate change curve during the corrosion process.

**Figure 8 materials-15-07023-f008:**
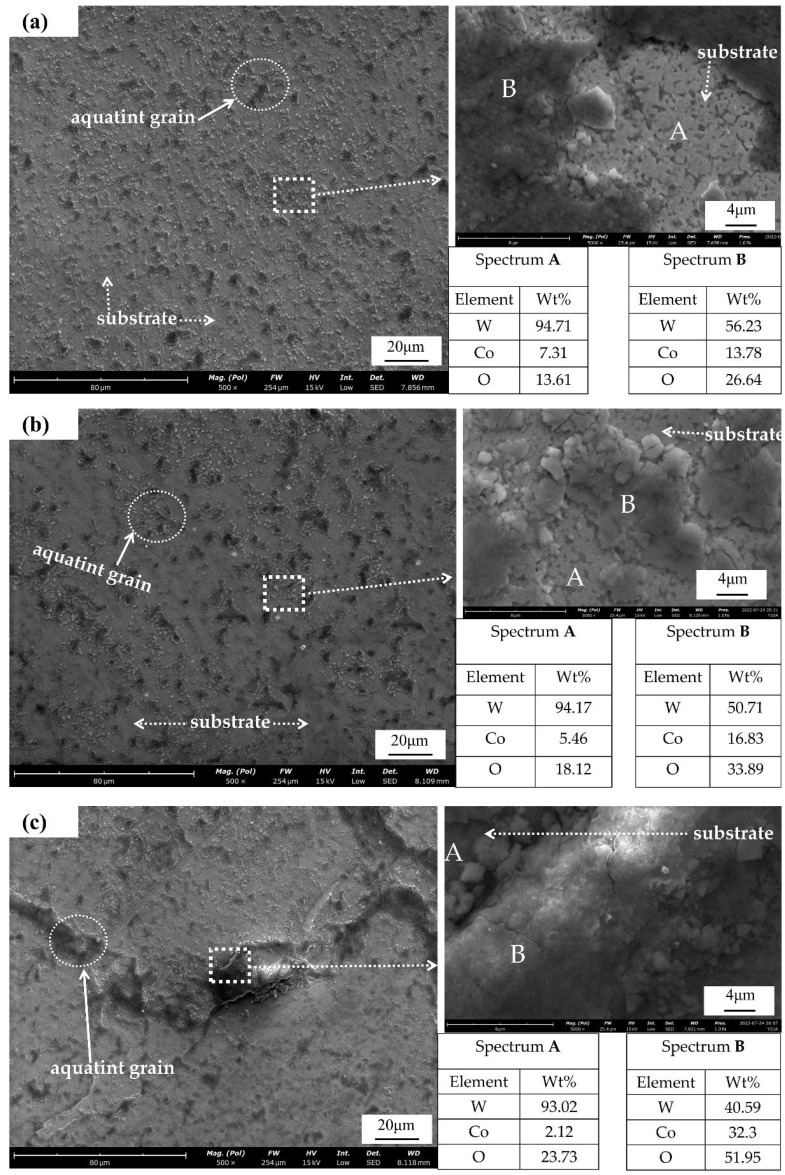
SEM and EDS elemental analysis of sample after (**a**) 12 h, (**b**) 24 h, and (**c**) 36 h of corrosion.

**Figure 9 materials-15-07023-f009:**
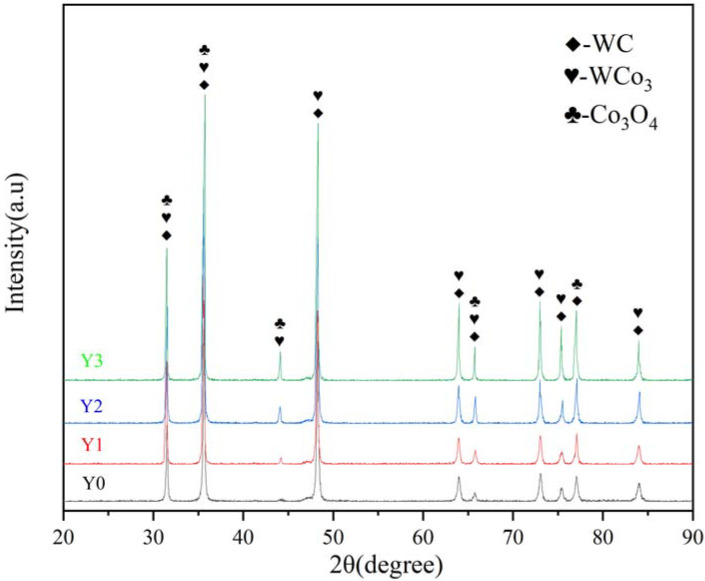
Surface XRD analysis of cemented carbide samples with different degrees of corrosion.

**Figure 12 materials-15-07023-f012:**
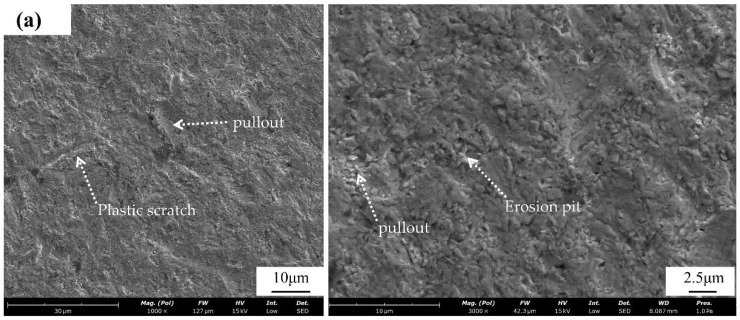
(**a**) Erosion of uncorroded SEM morphology sample; erosion of SEM morphology sample after (**b**) 12 h, (**c**) 24 h, and (**d**) 36 h of corrosion.

**Table 1 materials-15-07023-t001:** Main chemical composition of the mixed powder.

Chemical Composition				Physical Properties		
WC (wt%)	Co (wt%)	Ta (wt%)	Tb (wt%)	Bulk Density (g/cm^−3^)	Particle Size (μm)	Hall Flow Rate (s/50 g)
90.10 ± 0.05	9.85± 0.30	0.42	0.15	3.03	0.7	40.7

**Table 2 materials-15-07023-t002:** Composition of WC/Co cemented carbide sample.

Element	WC	Co	Ta	Tb
Wt%	90.1	9.56	0.55	0.32

**Table 3 materials-15-07023-t003:** Salt spray test parameters.

Temperature (°C)	Relative Humidity (%)	Salt Mist Pressure (kPa)	Salt Mist Deposition Rate (mL·h^−1^)	pH
25 ± 2	93 ± 3	98~112	1.0~2.0	3.5~4.2

**Table 4 materials-15-07023-t004:** Erosion experiment parameters.

Grain Size (μm)	Density (g·cm^−3^)	Hardness (GPa)	Erosion Angle	Erosion Times (min)	Erosion Pressure (Mpa)
200	2.6	8.5~9.8	60°	10	3.0 ± 0.2

**Table 6 materials-15-07023-t006:** Sample mass before and after erosion.

Sample	m_b_ (g)	m_a_ (g)
Y0	8.805	8.799
Y12	8.777	8.769
Y22	8.767	8.759
Y32	8.795	8.786

Note: m_b_: Weight before erosion, m_a_: weight after erosion.

## Data Availability

Not applicable.

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
