# Peer review of "Preparation of Cemented Carbide and Study of Copper-Accelerated Salt Spray Corrosion and Erosion Behavior"

_materials, 2022, doi:10.3390/ma15197023_

Round 1
Reviewer 1 Report
The paper has extensive English language and grammar mistakes. There are many Chinese in the figures and equations. Reviewers have no time to waste reviewing the raw paper that is not been carefully revised by the authors! The logic of the introduction section is confusing. Most discussions are not well supported by their results.
1. Give the full name of the abbreviations when first time show these abbreviations. Like CASS, MWD, WC-Co and etc.
2. The words in Figure 1 and 3 are Chinese. There are no indications of (a) and (b) in Fig.2.
3. It is surprising to see such a huge error for the concentration of NaCl in the salt spray test. “the solution using a concentration of 50 g / L ± 5 g / L of NaCl”
4. What are the results or main findings of Ref 11 and 12 (Line 48 to 51). You just listed the method they used to make the cemented carbide material, but you didn't tell us whether the method was good or not, and whether the material they made have any disadvantages or advantages. Describe the work of others in more detail. The same comment for the following introduction.
5. Check your English grammar, there are too many mistakes, and it’s hard for me to understand your words.
6. The discussion on the hollow spherical structure (line 165) in Fig.4 is not convincing. The SEM photos of crossed spherical powder is needed to support your discussion.
7. The EDS mapping on Fig.4 (b) is needed, otherwise, there is no reason for me to believe your discussion “WC grains are covered by Co phase…………”.
8. I can’t tell the different phases you described in Fig.5(a) and even in Fig.5 (b). “Figure 5 (a) Scanning electron microscope (FE-SEM, ZEISS) secondary electron mode scan of the carbide obtained after etching with solution (). It mainly contains the WC and cobalt phases (γ phase) of the alloy, WC phase, cobalt phase (γ phase) and η phase.”
9. There is a Chinese in Equation 2. Please check your paper carefully before submission.
Author Response
We have made detailed modifications as you suggested.Please see the attachment.

Reviewer 2 Report
Reviewer comments
The authors have presented a study on the Preparation of cemented carbide and a study of CASS and its erosion-corrosion behavior. They have examined the acid corrosion and erosion behavior of cemented carbide materials to provide a reference for the application of cemented carbide materials in the petrochemical industry. Overall, I found this article somewhat encouraging. I feel that there is very much detail in the observations, and statements are made with good supporting evidence; the article seems somewhat specific and very deep, even with the overall length of the paper. I recommend this study for publication after some minor corrections.
1. While written English can be followed, there is certainly room for improvement. Some of the sentence construction is challenging to follow, and a few examples make very little sense.
2. Authors should write the full form of “CASS” ….in..Title…..(some other information as well…..at least 2 lines in an introduction).
3. Since the authors have also performed the corrosion study in detail, hence the title should be modified…. Like “Preparation of cemented carbide materials and study of CASS and its erosion and corrosion behavior”
4. “the metal compounds of WCo3 and Co3O4” this is a corrosion product formed on the surface during salt fog exposure. It is not a metal compound.
5. The written language should be in English in Fig.1.
6. The author should add the schematic diagram of experiments in Fig. 3.
7. In table 2, W should be replaced by WC.
8. The caption should be written properly in table 4. Which should provide the full information on the parameters of erosion.
9. “SEM and EDS analysis of corrosion” should be replaced by SEM and EDS analysis of corroded WC-Co surface.
10. In figure 6… the “Rust” should be replaced by corrosion products since it is mainly for steel.
11. In table 5. Write the full form of M0 and M1 in the table as well.
12. Fig.7 and Fig. 11 should be in the form of bar plots with error bars.
13. “Analysis of erosion performance” should be replaced by erosion-corrosion behavior of WC-Co materials
14. What is it mean of purging morphology……
15. Authors should also perform the Raman along with XRD to confirm the formed corrosion products during the corrosion and erosion-corrosion experiments.
16. In table 6. Write the full form of Mb and Ma in the table as well.
17. Authors should also differentiate and illustrate the corrosion mechanisms and erosion-corrosion mechanisms in detail with the help of a schematic diagram.
18. The authors should give a very good reason and proper justification for the necessity of this work.
19. Authors should add some references “Failure Behavior of Cemented Tungsten Carbide Materials: A Case Study of Mining Drill Bits” and…” A comprehensive review on synergy effect between corrosion and wear of cemented tungsten carbide tool bits: A mechanistic approach” and ”Corrosion behavior of WC-Co tool bits in simulated (concrete, soil, and mine) solutions with and without chloride additions”

Author Response

(The authors gave the same response as above.)

Reviewer 3 Report
In this study, the authors gave us some knowledge about the preparation of cement carbide (WC-Co) and the study of erosion and corrosion properties in the application of making mud pulser (crude oil exploration). The work found several results of these properties as the purpose which authors have told, but the way author expresses ideas are not really alright. The problems can be listed as the following questions:
Introduction and sample preparation:
1. There are no sentences mentioned about CASS but the title, is “Copper accelerates acid salt spray”? Authors should consider adding some information about CASS.
2. Some Chinese letters in figures need to be replaced by English ones for reader to follow. (figure 1, 3 and in equation (2)).
3. Why did the authors include part 2.1 without the name of part 2?
4. Why in figure 3, it is 9h, 24h and 33h instead of 12h, 24h and 36h as the authors mentioned above? The authors label the samples in a very confusing manner and do not even utilize all these symbols.
Results and Discussion:
5. The sentence: “The more rounded the grains and the smoother the surface, the better the flowability of the powder mixture, and the more stable the molding in the high-speed pressing process.” Where is the ref that helps the Authors have this explanation?
6. from lines 190-194, the effect of adding some element, how can authors all this information? In what Refs?
7. where is γ and η phase in the figure (line197)?
8. The author has made measurements the coercive magnetic force and the magnetic saturation strength, so what do these measurements mean and how do they affect the material under consideration in this paper? what is 20KA/m value (line226)? And why do we need this value?
9. The equation (2), the parameter m1, why authors mentioned table 5 in this case? And moreover, why table 5 is mentioned nowhere else but this? Is it necessary to include table 5 in the article when its function is unclear? And why it is only one sample (Y33) in this table?
10. Why do the authors refer to measuring the samples in the corrosion rate measurement (figure 7) while there is only one line illustrated? Are the authors averaging the results of the sample?
11. in figure 9, XRD pattern, the sample Y0 is corresponding to carbide surface after salt spray corrosion for 0h, so, why is no change in phase from 0h to 36h? Does it mean that already have WC, WCo3 and Co3O4 phases from the initial sample? Authors should verify the standard WC, Co3W and Co3O4 XRD patterns, it could not be the same in almost peaks.
12. Finally, there are many paragraphs where the font is not the same, the authors need to consider editing, for example, section 3.1.2 is written straight without italic like 3.1.1, ...
13. Why the erosion rate is too low, Figure 11? some comparison with other studies should be provided?
Others:
Type of ball mill: horizontal/vertical/planetary ball mill should be provided. And line 89, the content in () should be rewritten to meaningful content. What is 90%, and What is 0.83 g/cm3 for powder or green compact sample?. Do you use any kind of inert gas for atmosphere control?.
Some unit should be corrected such as kA/m for KA/m, kA for KA in Ampere unit; MPa for Mpa; in table 4, "; or ()" is used for /, for example Grain size, μm or Grain size (μm) for Grain size/μm; kPa for Kpa and pH for PH in table 3.
What is the density of 0.83 g/cm3 in line 89, and 3.03 g/cm3 in table 1 and sintered sample?
A missing content () of etching solution is in line 196. And comparison not in the same unit in line 213.
Author Response

(The authors gave the same response as above.)

Round 2
Reviewer 1 Report
The results and conclusions presented in the manuscript is fine. The manuscript could be accepted after adding the error bar for the corrosion rate in Fig.7 and moderate English changing.
Reviewer 3 Report
The authors have generally corrected the errors mentioned in the previous review. But here are some additional suggestions:
- Authors might consider including an explanation or re-editing the XRD in order to clarify the XRD results. The indexing peaks of WC, WCo3 and Co3O4 are not the same. The authors should ask an XRD expert for this.
- If the authors already have an explanation for why only sample Y33 appears in Table 5, authors might consider including it in the manuscript. Similar to figure 7, with an explanation as to why only one line is depicted.
- The author should consider further explaining the influence of the coercive magnetic force and the magnetic saturation strength on this work in the introduction.
- Authors should consider adding a ref to this explanation: "A small amount of added Ta can refine the WC grains, but too much Ta can lead to a decrease in alkali corrosion resistance and toughness of the alloy. amount of element Tb is added to improve the coercivity during sintering. element W will form the main alloy component WC with C, which is used to Regulate the cobalt phase (γ phase) composition, crystalline shape, and grain size of the alloy."
